# HIV-Differentiated Metabolite N-Acetyl-L-Alanine Dysregulates Human Natural Killer Cell Responses to *Mycobacterium tuberculosis* Infection

**DOI:** 10.3390/ijms24087267

**Published:** 2023-04-14

**Authors:** Baojun Yang, Tanmoy Mukherjee, Rajesh Radhakrishnan, Padmaja Paidipally, Danish Ansari, Sahana John, Ramakrishna Vankayalapati, Deepak Tripathi, Guohua Yi

**Affiliations:** 1Department of Medicine, The University of Texas at Tyler School of Medicine, Tyler, TX 75708, USA; 2Center for Biomedical Research, The University of Texas at Tyler School of Medicine, Tyler, TX 75708, USA; 3Department of Cellular and Molecular Biology, The University of Texas Health Science Center at Tyler, Tyler, TX 75708, USA

**Keywords:** metabolite, HIV–*Mtb* co-infection, N-acetyl-L-alanine, natural killer cells, interferon-γ, *Mtb* latent infection (LTBI)

## Abstract

*Mycobacterium tuberculosis* (*Mtb*) has latently infected over two billion people worldwide (LTBI) and caused ~1.6 million deaths in 2021. Human immunodeficiency virus (HIV) co-infection with Mtb will affect the Mtb progression and increase the risk of developing active tuberculosis by 10–20 times compared with HIV- LTBI+ patients. It is crucial to understand how HIV can dysregulate immune responses in LTBI+ individuals. Plasma samples collected from healthy and HIV-infected individuals were investigated using liquid chromatography–mass spectrometry (LC-MS), and the metabolic data were analyzed using the online platform Metabo-Analyst. ELISA, surface and intracellular staining, flow cytometry, and quantitative reverse-transcription PCR (qRT-PCR) were performed using standard procedures to determine the surface markers, cytokines, and other signaling molecule expressions. Seahorse extra-cellular flux assays were used to measure mitochondrial oxidative phosphorylation and glycolysis. Six metabolites were significantly less abundant, and two were significantly higher in abundance in HIV+ individuals compared with healthy donors. One of the HIV-upregulated metabolites, N-acetyl-L-alanine (ALA), inhibits pro-inflammatory cytokine IFN-γ production by the NK cells of LTBI+ individuals. ALA inhibits the glycolysis of LTBI+ individuals’ NK cells in response to *Mtb.* Our findings demonstrate that HIV infection enhances plasma ALA levels to inhibit NK-cell-mediated immune responses to *Mtb* infection, offering a new understanding of the HIV–*Mtb* interaction and providing insights into the implication of nutrition intervention and therapy for HIV–*Mtb* co-infected patients.

## 1. Introduction

*Mycobacterium tuberculosis* (*Mtb*), the causative pathogen for tuberculosis, is responsible for ~2 billion latent infections (LTBIs) globally, with ~1.6 million deaths in 2021. Among these *Mtb*-infected individuals, 6.7% are co-infected with human immunodeficiency virus (HIV) [1]. Even though the *Mtb* latent infection is asymptomatic, once co-infected with HIV, the risk of developing active TB in LTBI patients is 10–20 times more than in HIV-uninfected LTBI+ individuals [2]. To develop better vaccines and treatment methods, it is important to understand the HIV-mediated dysregulation of immune responses in LTBI+ individuals.

Cellular metabolism plays a crucial role in regulating human immune responses to pathogenic infections [3,4]. The accumulation of specific metabolites from the pathogen-infected cells can function as epigenetic modifiers to immune cells and alter the epigenetic landscape of some metabolically important enzymes [5], and it leads to changes in immune cell homeostasis and/or functional changes in the affected immune cells [6]. Eventually, it enhances the pathogen’s growth and disease progression. During HIV infection, the metabolism of host cells will be skewed to viral survival and replication [7,8]. HIV-infected macrophages are known to be metabolically altered with the characteristics of mitochondrial fusion, lipid accumulation, and reduced mitochondrial ATP production [9]. HIV-induced metabolites, such as glucose and some amino acids and their intermediate products, have been reported to significantly impact the function of the immune system. For instance, glucose uptake is essential for the activation of CD4+ T cells and the pro-inflammatory cytokine production in myeloid cells during HIV infection [10,11,12]. Mechanistically, it regulates CD4+ T cells or myeloid cells via the tricarboxylic acid cycle (TCA cycle) [13].

However, little is known about how HIV-driven host metabolite changes can dysregulate the immune system and control *Mtb* immunopathogenesis, which is an important question that needs to be answered in terms of controlling *Mtb* progression in HIV–*Mtb* co-infected patients. In this study, we performed a metabolomic comparison using the plasma from healthy donors and HIV-infected patients (with or without antiretroviral therapy (ART)). One of the metabolites, N-acetyl-L-alanine (ALA) was more abundant in HIV+ plasma than in healthy donor samples. ALA inhibited pro-inflammatory cytokine IFN-γ production by the NK cells of LTBI+ individuals. We also found that ALA inhibited the glycolysis of LTBI+ individuals’ NK cells in response to *Mtb.* Our findings demonstrate that HIV infection enhances plasma ALA levels to inhibit NK-cell-mediated immune responses to *Mtb* infection.

## 2. Results

### 2.1. Metabolic Profiles of HIV-Positive Patient Plasma

To characterize the HIV patient-specific metabolic landscape, we performed liquid chromatography–mass spectrometry (LC-MS)-based metabolic profiling in plasma samples from HIV-positive patients (both treatment-naive and ART-treated) and healthy donors. We performed supervised partial least-square discriminant analysis (PLS-DA) of metabolome profiles and plotted the two principal components explaining the highest magnitude, as shown in Figure 1A. We observed that the plasma metabolome landscape of HIV patients (both treatment-naïve and ART-treated) was distinct from healthy controls, while the profiles of treatment-naïve and ART-treated patients were similar (Figure 1A). To identify the metabolites that are altered in HIV infection, we performed differential enrichment analysis and identified 60 metabolites that showed altered abundance in patients compared with healthy donors at a false discovery rate (FDR) of <0.05. We performed hierarchical clustering on differentially enriched metabolites (Figure 1B). As expected, healthy donors and HIV patients formed distinct clusters, which is consistent with the PLS-DA results (Figure 1B).

However, treatment-naïve and ART-treated patients formed a single cluster, underscoring similarities in the metabolome profiles of both treatment groups. The most significant metabolites altered between healthy donors and patients were selected using the following criteria: (1) the FDR value ranks beyond the first 60; (2) the VIP score is >1 based on PLS-DA analysis; and (3) the fold change is >1.5 (HIV/healthy donors for upregulated metabolite and healthy/HIV+ donors for downregulated metabolites). Finally, we found that two metabolites, N-acetyl-L-alanine (ALA) and glycine, were upregulated in HIV-positive plasma samples, and six metabolites were downregulated, namely acetoacetate, glutarylcarnitine, lumichrome, O-succinylcarnitine, theodromine, and thymidine monophosphate, respectively (Figure 1C).

### 2.2. ALA Inhibits IFN-γ and TNF-α Secretion by γ-Irradiated Mtb (γMtb)-Stimulated PBMCs of LTBI+ Donors

Pro-inflammatory cytokines IFN-γ, TNF-α, IL-17A, and IL-1β are known to play important roles in controlling *Mtb* infection [14], and anti-inflammatory cytokine IL-10 plays an important role in the reactivation of TB [15]. We investigated whether the above-identified metabolites have any effect on the production of these cytokines by PBMCs obtained from LTBI+ donors. As shown in Appendix A, we performed an LDH assay to select the optimal concentrations of the metabolites for in vitro experiments. We cultured PBMCs from healthy LTBI+ donors with or without γ*Mtb,* as mentioned in the Methods section. Some of the γ*Mtb*-cultured PBMCs were cultured with the metabolites. After 72 h, culture supernatants were collected, and cytokine levels were determined using ELISA. ALA (3 µM concentration) significantly inhibited γ*Mtb*-stimulated IFN-γ, TNF-α, and IL-17 production by the PBMCs of LTBI+ individuals (Figure 2A–C). In contrast, other metabolites had no effects on the production of IFN-γ, TNF-α, and IL-17 of LTBI+ PBMCs (Figure 2A–C). None of the metabolites had any effect on the production of IL-1β, IL-10, and IL-13 (Figure 2D–F).

### 2.3. ALA Inhibits IFN-γ Secretion of NK Cells

We determined the effects of ALA on the expansion of various immune cell populations in the above-cultured cells. We found that ALA triggered the population expansion of the classic monocytes (CD14+CD16-), while it did not affect the expansion of other immune cells and their subpopulations (Appendix A). To determine the cellular source for the IFN-γ and TNF-α, various cell populations in the above-cultured cells were sorted (cell purity is shown in Appendix A), and a quantitative RT-PCR (qRT-PCR) was performed to determine the IFN-γ and TNF-α transcription levels. ALA significantly inhibited IFN-γ gene expression by NK cells and CD8+ T cells (Figure 3A). In contrast, TNF-α gene expression by the above immune cell populations was not affected by ALA (Figure 3B). We further confirmed the above findings at the protein level by performing intracellular staining on the above immune cell populations and found that ALA inhibited IFN-γ production by NK cells in response to γ*Mtb* (Figure 4C,D). In contrast, ALA had no effect on IFN-γ production by CD8+ and CD4+ cells (Figure 4A,B,D).

### 2.4. ALA Inhibits Nuclear Factor Kappa B (NF-κB), Activator Protein-1 (AP1), and Antimicrobial Peptide Expression by γMtb-Cultured NK Cells

We cultured PBMCs from healthy LTBI+ donors with or without γ*Mtb* stimulation, as mentioned in the Methods section. Some of the γ*Mtb*-cultured PBMCs were cultured with 3 µM ALA. After 48 h, NK cells were sorted, and qPCR was performed to determine the expression of 22 transcription factors and signaling molecules. Among these, ALA significantly inhibited the gene expression of NF-κB, AP1, and antimicrobial peptides GZMA and GZMB in γ*Mtb*-cultured NK cells. In contrast, SATA4 expression in γ*Mtb*-cultured NK cells was significantly upregulated by ALA. (Figure 5)

### 2.5. ALA Does Not Alter Cell Death Molecules in γMtb-Cultured NK Cells

In the above-cultured NK cells, we also determined the expression of 11 key genes involved in various death pathways (i.e., autophagy, apoptosis, pyroptosis, necroptosis, and ferroptosis). We found that ALA did not affect the cell death pathways tested in this study, while Atg3 expression (autophagy-related gene) alone was significantly upregulated when compared to the untreated and γ*Mtb*-stimulated cells (*p* = 0.0023) (Figure 6).

### 2.6. ALA Restricts the Bioenergetic Machinery in NK Cells

Metabolic switch to a glycolytic/energetic phenotype supports diverse NK cell functions [16,17]. We determined whether ALA treatment affects the metabolic state of γ*Mtb*-cultured NK cells. We performed a metabolic flux assay (as mentioned in the Methods section) to detect changes in the mitochondrial oxygen consumption rate (OCR) and the rate of extracellular acidification (ECAR) as measures of oxidative phosphorylation and glycolysis, respectively.

Freshly isolated PBMCs from LTBI donors (*n* = 3) were cultured in the presence of γMtb. Some of the γMtb-cultured PBMCs were also supplemented with ALA (3 µM). After 48 h, NK cells were isolated from the cultured PBMCs, and a metabolic flux assay was performed using a seahorse analyzer, as mentioned in the Methods section. As shown in Figure 7A,B, various parameters, namely basal respiration, ATP production, and the spare respiratory capacity of oxidative phosphorylation, were significantly reduced in the NK cells from ALA-alone-treated PBMCs and γMtb-alone-cultured PBMCs than control PBMCs. However, we observed a significant marginal reduction in the basal ATP production rate in NK cells from the PBMCs cultured with γMtb and ALA together than γMtb alone. Surprisingly, we saw pronounced changes in glycolytic parameters as well in NK cells from ALA-alone-treated PBMCs and γMtb-alone-cultured PBMCs compared with control PBMCs (Figure 7C,D). Interestingly, we found ALA treatment further significantly reduced basal glycolysis, glycolytic capacity, and glycolytic reserve in NK cells from γMtb-cultured PBMCs compared with γMtb-alone-cultured PBMCs (Figure 7B). Herein, we observed that ALA treatment significantly suppressed OXPHOS and glycolysis in NK cells, which suggests that a higher level of ALA in HIV patients can induce quiescent phenotypes in NK cells.

## 3. Discussion

Immunometabolism plays a central role in host–*Mtb* interactions and controls the infection outcomes [18,19,20]. It is not known whether metabolic changes during HIV infection alter immune responses to *Mtb* infection. In the current study, we found that HIV infection altered plasma metabolic profiles. Among the various elevated metabolites, ALA significantly inhibited NK-cell-mediated immune responses to *Mtb* infection. We also found that ALA inhibited the expression of transcription factors NF-κB, AP1, GZMA, and GZMB, which are important in the activation and antimicrobial activity of NK cells.

Activated NK cells produce IFN-γ, which activates macrophages to kill intracellular organisms [21,22,23]. It has been demonstrated that human NK cells have the potential to contribute to both innate and adaptive immune responses to *Mtb* [24,25,26,27,28,29]. NK cells can lyse *Mtb*-infected monocytes and alveolar macrophages through the NKp46 receptor and NKG2D [25,28], and NK cells contribute to the capacity of CD8+ T cells to produce IFN-γ and to lyse *Mtb*-infected monocytes [26]. During chronic HIV infection, an abnormal, dysfunctional CD56neg NK cell subset expands, and potentially protective NK cell responses are depressed [30,31,32]. However, limited information is available on NK cell response to pathogens, including *Mtb*, especially in HIV+LTBI+ individuals.

HIV infection induces significant immunometabolic changes in the host [33,34,35]. NK cell cytotoxicity and cytokine production depend on their metabolism [36], and altered metabolism is linked to NK cell dysfunction [37]. No information is available about the metabolic requirements of NK cells during *Mtb* and/or HIV infection. Metabolomics provides a versatile tool to study the host immune responses to pathogen infections because metabolism offers a source of energy required for immune cell function. During HIV and *Mtb* infection/co-infection, immune cell activation and inflammation have also been shown to correlate with metabolic changes in the immune cells [38,39,40,41]. However, the mechanisms of how specific metabolites affect the immune responses to HIV/*Mtb* infections remain elusive. Our study shows that HIV-induced metabolite ALA can inhibit IFN-γ production and antimicrobial peptide expression in NK cells and inhibits NK cell glycolysis in response to *Mtb*. As glycolysis is essential to maintain cell viability and inflammation activity [42,43], depressed glycolysis causes NK cell autophagy and a reduction in IFN-γ production, as demonstrated in this study. IFN-γ is one of the major cytokines that can limit *Mtb* growth; thus, increased ALA levels during HIV infection can enhance *Mtb* growth and disease progression. It is worth noting that we identified the commonly important metabolites between the treatment-naïve and ART-treated samples when performing the metabolomic analysis. This strategy may help researchers in finding potential nutrition interventions/therapies that are suitable for both ART-treated and untreated HIV–*Mtb*-co-infected patients.

Amino acids participate in energy production during cell metabolism, and some amino acids are involved in oxidative stress and redox signaling during HIV and *Mtb* infection [38,39]. These physiological activities are tightly coupled with immune activation, as revealed by a previous study indicating that alanine is essential for CD4+ T-cell activation [44]. ALA is the derivative of L-alanine. It can be produced via the direct synthesis of N-acetyltransferases or the proteolytic degradation of N-acetylated proteins by hydrolases, such as aminoacylase I [45]. In our study, ALA upregulation suggests a decrease in the non-acetylated L-alanine. Consequently, the redox-sensitive transcription factors such as NF-κB and AP-1 are downregulated via the redox signaling pathway, and this is also the case in the LTBI NK cells (Figure 5). In another scenario, ALA may be able to compete with non-acetylated L-alanine to bind to the same nutrition receptor, leading to a decrease in the L-alanine uptake and other similar consequences.

NK cells have been reported to use glycolysis and oxidative phosphorylation (Oxphos) pathways to provide energy for various physiological activities, such as activation and proliferation [46,47]. We found that the glycolysis of γ*Mtb*-activated NK cells was significantly reduced, and this may explain the decline in IFN-γ production due to the growing consensus that glycolysis is critical for IFN-γ production by NK cells [48].

Immune cell metabolism plays a vital role in shaping immune responses to pathogen infection. Effector immune cells are believed to upregulate glycolysis to enable a quicker turnover of ATP, essentially switching to a state of aerobic glycolysis to meet the urgent demand for a mounting response to pathogenic challenges in the form of increased proliferation, production of cytokines, and other cytotoxic capabilities [49]. This enhancement usually occurs through the upregulation of glycolytic enzymes and the upregulation of surface nutrient transporters such as CD71, CD98, and Glut1 [50]. Several transcription factors are also involved in orchestrating metabolic rewiring, NF-κB (nuclear factor kappa-light-chain-enhancer of activated B cells), HIF-1α (hypoxia-inducible factor-1α), c-Myc, Akt, and mTOR (mechanistic target of rapamycin) are all known to differentially regulate the glycolytic gene expression landscape upon stimulation [51]. Apart from being a quicker source of energy, glycolysis also fuels the pentose phosphate pathway, which increases the availability of PPP intermediates (ribose-5-phosphate and NADPH) essential for proliferation and effector functions [52]. Conversely, glycolytic end products can also be shunted into the TCA cycle as acetyl-CoA, NADH, and FADH2 to further support OXPHOS, essentially supporting an energetic phenotype [53]. Recent studies have shown that NK cells lacking in lactate dehydrogenase A lose their tumorigenicity and antiviral function, suggesting an indispensable role of glycolysis [54,55,56].

Collectively, we conclude that the mechanism involving ALA is as follows: ALA functions as a downregulator of NK cell glycolysis, and then the downregulated glycolysis pathway will result in NK cell autophagy and a reduction in IFN-γ production, as shown in Figure 8. We hypothesize that HIV can upregulate ALA and thus stimulate *Mtb* growth during HIV–*Mtb* co-infection. Our study offers a new understanding of the HIV–*Mtb* interaction and provides insights into the implication of nutrition intervention and therapy for HIV–*Mtb* co-infected patients. This will be further investigated in our future work.

## 4. Materials and Methods

### 4.1. Human Study Sample Collection

All healthy donor samples (5 samples for each experiment), HIV-positive plasma samples (8 samples), and LTBI+ blood samples (5–6 samples for each experiment) were collected according to the protocols approved by, respectively, the Institutional Review Boards of Texas Tech University Health Sciences Center at El Paso and the University of University Health Science Center at Tyler. All the participants in this study provided written informed consent.

HIV+ patients were recruited based on the inclusion/exclusion criteria: the HIV+ individual must be 20–60 years old, with a positive AMPLICOR HIV-1 Monitor test [57] (either with or without ART treatment), and without Mtb infection, diabetes, pregnancy, cancer, autoimmune diseases, or any other immunosuppressive conditions. All the HIV+ patient blood samples were collected in 2014. Detailed information on HIV probands is shown in Appendix A.

The inclusion/exclusion criteria of LTBI+ individuals were based on the QuantiFERON-TB Gold Plus (QFN) test [58]. Those who were QFN- were considered LTBI-, and QFN+ donors were evaluated for TB using chest radiography and clinical evaluation per the guidelines [59]. Active TB patients were excluded. Similarly, those who had other comorbidities such as HIV infection, diabetes, cancers, and other immunosuppressive diseases were excluded.

HIV-positive peripheral blood samples were collected into tubes containing sodium heparin and centrifuged at 8000× *g* for 10 min at 4 °C for 15 min, and the plasma samples were pipetted out and stored at −80 °C until use.

Blood was collected at the Pathology Laboratory of the University of Texas Health Science Center at Tyler, and PBMCs were isolated using Ficoll–Paque (Fisher Scientific Inc., Waltham, MA, USA) density gradient centrifugation as per the manufacturing instructions.

### 4.2. Antibodies and Flow Cytometry

The antibodies used for this study’s surface and intracellular staining were purchased from Biolegend Inc., San Diego, CA, USA. These fluorescence-labeled antibodies were used for staining different panels: APC-Cy7-CD3 (clone HIT3a), PE-CD4 (A161A1), PE-Cy7-CD45 (H130), PE-Dazzle 594-CD56 (NCAM) (HCD56), BV605-CD8 (SK1), BV421-FoxP3 (206D), APC-CD25 (M-A251), APC-TNFα (MAb11), BV421-IFNγ (4S.B3), BV510-KLRG1 (2F1/KLRG1), APC-CD27 (M-T271), FITC-CD4 (SK3), BV711-CD25 (BC96), PerCP-Cy5.5-PD-1 (EH12), PE-Cy7-CD8 (SK1), BV421-CCR7 (G043H7), BV605-CD56 (HCD56), PE-CD62L (DREG-56), PE-Cy5-CD19 (HIB19), BV711-CD16 (3G8), FITC-CD14 (HCD14), BV421-CD14 (HCD14), BV605-CD11c (3.9), BV605-CD11b (ICRF44), PB-CD45 (2D1), and APC-CD40L (24–31). The isotype antibodies used for this study were as follows (the same clones were chosen as the above fluorescence-labeled antibodies): APC mouse IgG1, BV421 mouse IgG1, PerCP/Cy5.5 mouse IgG1, BV605 mouse IgG1, PE/Dazzle 594 mouse IgG1, BV711 mouse IgG1, PE Rat IgG2b, BV510 mouse IgG2a, APC mouse IgG1, APC-Cy7 mouse IgG1, PE-Cy7 mouse IgG1, FITC mouse IgG1, and PB mouse IgG1. For surface staining, the cells were stained using different panels of fluorescence-labeled antibodies for 30 min on ice, and the stained cells were then washed in a FACS buffer (2% fetal calf serum (FCS) in PBS), and resuspended in a 500 µL FACS buffer. For intracellular staining, the surface-stained cells were fixed for 30 min at room temperature and intracellularly stained with BV421-IFNγ (4S.B3) in 1x permeabilization buffer for 20 min at room temperature using an Intracellular Fixation and Permeabilization Buffer Set (eBioscience™, San Diego, CA, USA; 88-8824-00). The stained cells were collected using Attune NXT (Thermo Fisher Scientific, St. Bend, OR, USA), and the data were analyzed with FlowJo (Tree Star, Ashland, OR, USA). Dead cells were removed using both forward and side scatter gating.

### 4.3. PBMC Treatments with Various Metabolites

Briefly, 2 million PBMCs in each well of a 12-well plate were stimulated with 10 µg/mL γ-irradiated *Mtb* (γ*Mtb*) and immediately followed by treatments with different concentrations of various metabolites. The unstimulated cells and stimulated but untreated cells served as negative controls. After 72 h, the cells were collected for surface and intracellular staining, and the supernatants were collected for ELISA to test the cytokine expressions.

### 4.4. Quantitative Reverse-Transcription PCR (qRT-PCR)

The γ*Mtb*-stimulated, ALA-treated, and untreated PBMCs were collected, and the RNA was extracted using a TRIzol reagent (Invitrogen, Waltham, MA, USA), as recommended by the manufacturer.

The mRNA transcription levels of the NK cell signaling molecules and the death pathways’ molecules were measured via qRT-PCR using β-actin as an internal control with specific primer sets (Integrated DNA Technologies) (see Appendix A in the Appendix A).

### 4.5. Extracellular Flux Measurement

PBMCs from LTBI+ healthy donors were plated in 12-well plates at a concentration of ~5 × 10^6^ cells/well. The cells were treated with N-acetyl-L-alanine or γ*Mtb* or both, along with no treatment for up to 48 h. After 48 h, NK cells were isolated from the respective wells using an NK cell isolation kit (Miltenyi Biotec, Tokyo, Japan; Cat: 130-092-657) following standard protocol and plated at a concentration of 2 × 10^5^ cells per well using a seahorse XFe96 assay plate in seahorse XF DMEM media (Agilent, Santa Clara, CA, USA; 103575-100) supplemented with 1 mM pyruvate, 2 mM glutamine, and 10 mM of glucose. OXPHOS measurements were performed in a Seahorse Xfe96 Analyzer, using a mito-stress test kit (Agilent; 103015-100). The measurement of OCR (oxygen consumption rate) was carried out after the subsequent addition of 1.5 μM oligomycin, 1 μM FCCP (carbonyl cyanide-4 trifluoromethoxy phenylhydrazone), and 0.5 μM rotenone/antimycin A (Rot/AA). Basal respiration was measured as the OCR after subtracting the non-mitochondrial respiration rate obtained after adding Rot/AA, spare respiratory capacity was measured as the highest respiration obtained compared with basal respiration after adding FCCP, and ATP-coupled respiration was defined as the OCR value affected by the addition of oligomycin. Glycolytic parameters were measured using a glycolysis stress test kit (Agilent; 103020-100); the measurement of ECAR (extracellular acidification rate) was carried out after the sequential addition of 1 mM glucose, 1.5 μM oligomycin, and 5 mM 2-deoxyglucose(2-DG). Basal glycolysis was measured as the resting ECAR value after the addition of glucose, while glycolytic capacity was defined as the maximum ECAR after the addition of oligomycin, and the glycolytic reserve was defined as the difference between the basal and maximum glycolytic capacity. Wave Desktop 2.6 software (Agilent) was used for the data analysis.

### 4.6. Metabolomics

Plasma samples were collected from healthy and HIV-infected individuals at Texas Tech University Health Science Center, El Paso. The plasma samples were analyzed at the metabolomic core facility at the Children’s Medical Center Research Institute at UT Southwestern (Dallas, TX, USA) using liquid chromatography–mass spectrometry (LC-MS). A triple-quadrupole mass spectrometer was used in MRM mode for the analysis, with two different dilutions for the samples, including four different retention times and three quality control samples. Further annotation of peaks was carried out using a proprietary database. The data matrix was statistically arranged using Metabo-Analyst (https://metaboanalyst.ca (accessed on 15 December 2022)), an online platform for reading metabolomic data using default parameters.

### 4.7. ELISA and LDH Assay

All ELISA kits were purchased from Thermo Fisher Scientific Inc, CA. The supernatants were collected from the PBMC culture after 72 h treatments. The ELISA procedures to detect IFN-γ, TNF-α, IL-1β, IL-4, IL-13, and IL-17A were performed according to the manufacturer’s protocols. A colorimetric CyQUANT lactate dehydrogenase (LDH) assay (Thermo Fisher Scientific Inc., Waltham, MA, USA) was performed to determine the LDH activity in culture supernatants of PBMCs.

### 4.8. Statistical Analysis

Each treatment was triplicated (qRT-PCR) or duplicated (all other experiments); each experiment was independently and reproducibly repeated two times, and representative results are presented. Power analysis was performed to determine the sample size to ensure biological significance. The data were analyzed using GraphPad Prism 9.0 software. A paired student’s *t*-test was used to analyze the difference between the treated and untreated samples from the same donor. Statistical significance was defined as * *p* ≤ 0.05, ** *p* ≤ 0.01, and *** *p* ≤ 0.001.

## Figures and Tables

**Figure 1 ijms-24-07267-f001:**
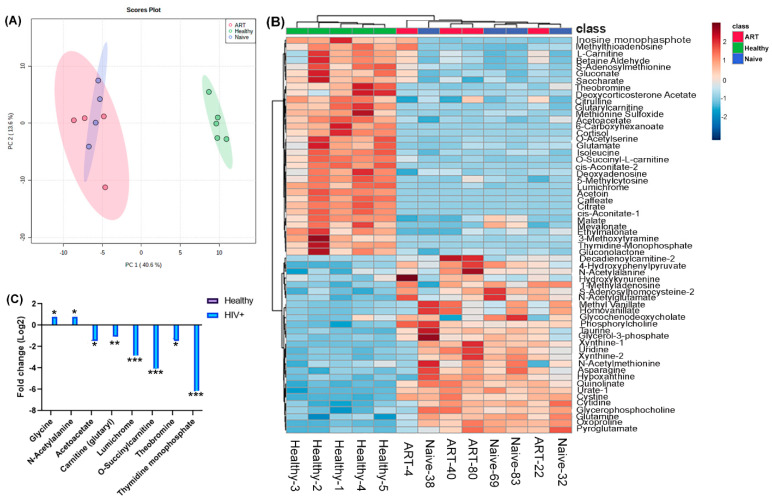
Plasma metabolic profiles of HIV+ patients (treatment-naïve and ART-treated): (**A**) Scatter plot showing partial least-square discriminant analysis (PLS-DA) of plasma metabolomic profiles. The two principal components explaining the highest variance were plotted on *X* and *Y* axis; (**B**) heatmap shows the top 60 differentially abundant metabolites identified in a comparison between healthy donors and HIV patients at 5% FDR; (**C**) fold change in abundance of 6 selected metabolites computed from metabolome profiles in a comparison between HIV patients and healthy donors. The bars represent log2 fold change in HIV patients compared with healthy donors. The asterisks *, **, and *** denote FDR < 0.05, FDR < 0.01, and FDR < 0.001, respectively.

**Figure 2 ijms-24-07267-f002:**
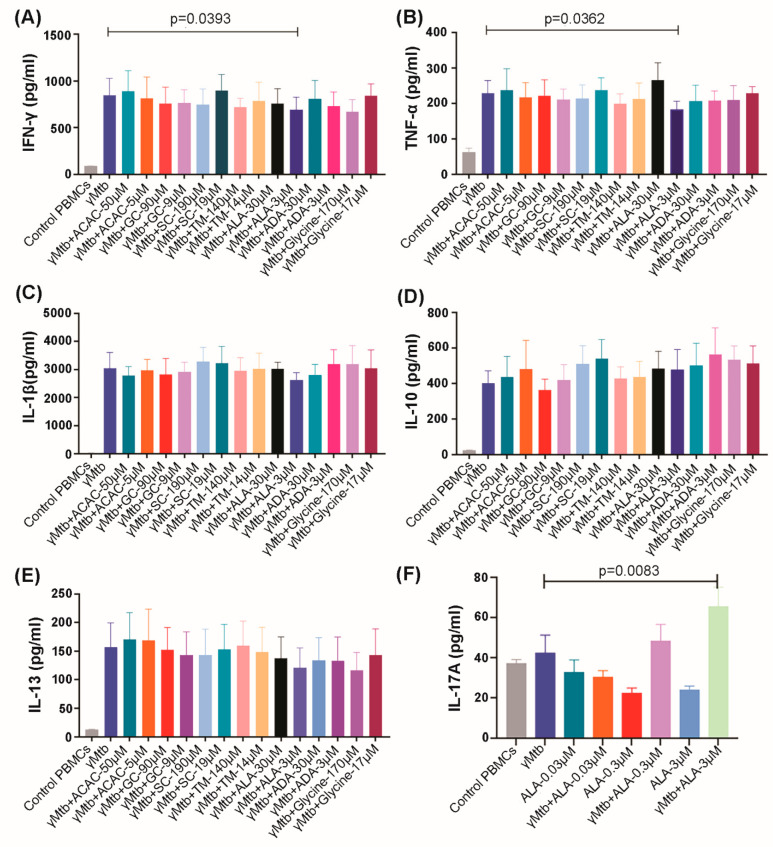
Effect of metabolites on cytokine production by PBMCs of LTBI+ donors. PBMCs of healthy LTBI+ donors were cultured with or without γ*Mtb* (10 μg/mL) and in the presence or absence of different concentrations of ALA and other metabolites, as mentioned in the Methods section. After 72 h, culture supernatants were collected, and cytokine levels were measured using ELISA: (**A**) IFN-γ, (**B**) TNF-α, (**C**) IL-1β, (**D**) IL-10, (**E**) IL-13, and (**F**) IL-17A. In (**A**–**E**), six donors were collected, while in (**F**), five donors were collected. Paired *t*-tests were used to compare the differences between untreated and treated PBMCs from the same samples. The mean values and SDs are shown, and the significant *p* values are shown (*p* < 0.05).

**Figure 3 ijms-24-07267-f003:**
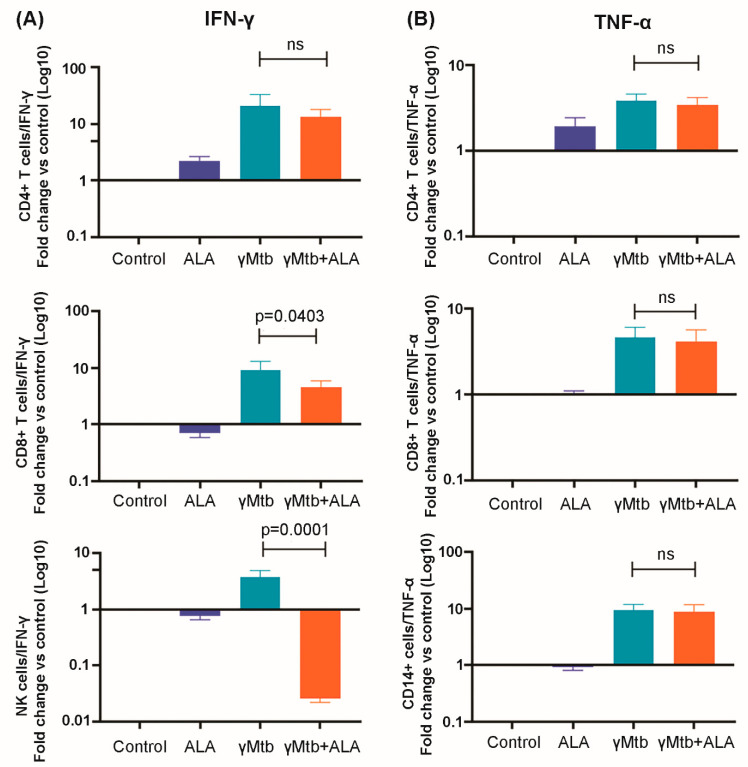
IFN-γ and TNF-α gene expression profile of various immune cells in γ*Mtb*-cultured PBMCs of LTBI+ donors. PBMCs of healthy LTBI+ donors were cultured with or without γ*Mtb* (10 μg/mL) and in the presence or absence of ALA, as mentioned in the Methods section. After 48 h, various immune cells were isolated via flow sorting, RNA was collected, and real-time PCR analysis was performed to determine IFN-γ and TNF-α gene expression: (**A**) IFN-γ transcription levels in different cell types; (**B**) TNF-α transcription levels in different cell types. PBMCs from five LTBI+ donors were used for the study. Paired *t*-tests were used to compare the differences between untreated and treated PBMCs from the same samples. The significant *p* values are shown (*p* < 0.05). ns: not significant.

**Figure 4 ijms-24-07267-f004:**
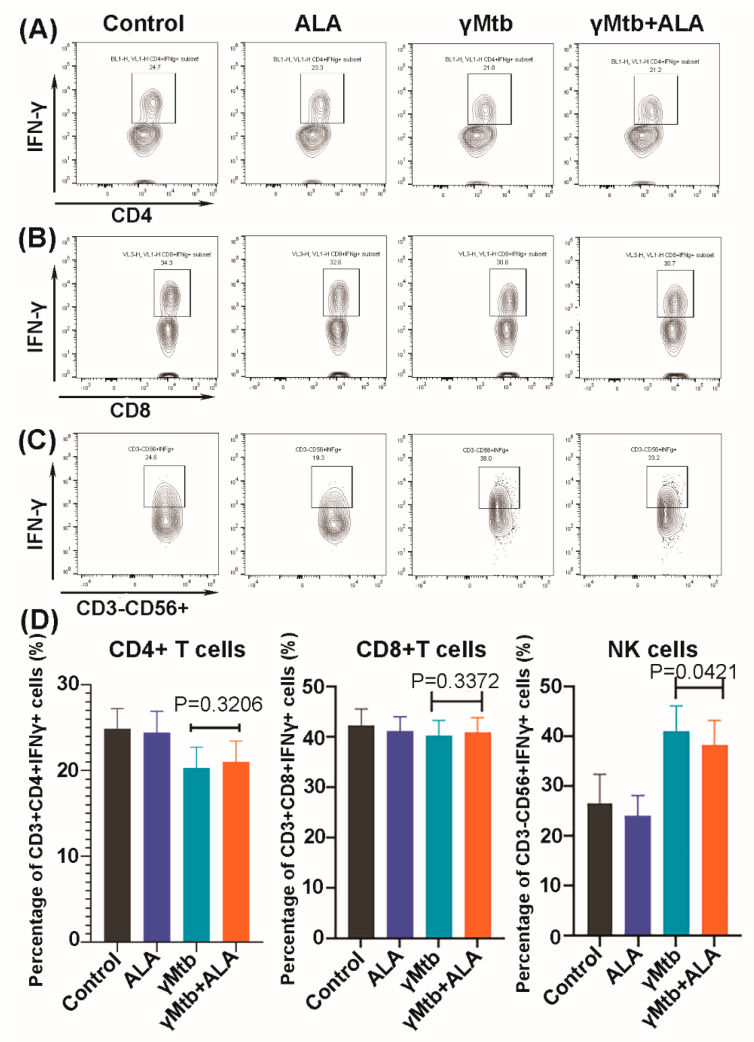
IFN-γ levels of various immune cells in γ*Mtb*-cultured PBMCs of LTBI+ donors. PBMCs of healthy LTBI+ donors were cultured with or without γ*Mtb* (10 μg/mL) and in the presence or absence of ALA, as mentioned in the Methods section. After 48 h, intracellular staining was performed to determine IFN-γ levels of various immune cell populations: (**A**–**C**) IFN-γ staining of a representative donor PBMC; the top, middle and bottom panels represent CD4+ T cells (**A**), CD8+ T cells (**B**), and NK cells (**C**), respectively. The percentages of IFN-γ-positive cells are shown; (**D**) collective summary of IFN-γ-positive CD4+, CD8+, and CD56+ cells of five donors. Paired *t*-tests were used to compare the differences between untreated and treated PBMCs from the same samples. The significant *p* values are shown in the figure.

**Figure 5 ijms-24-07267-f005:**
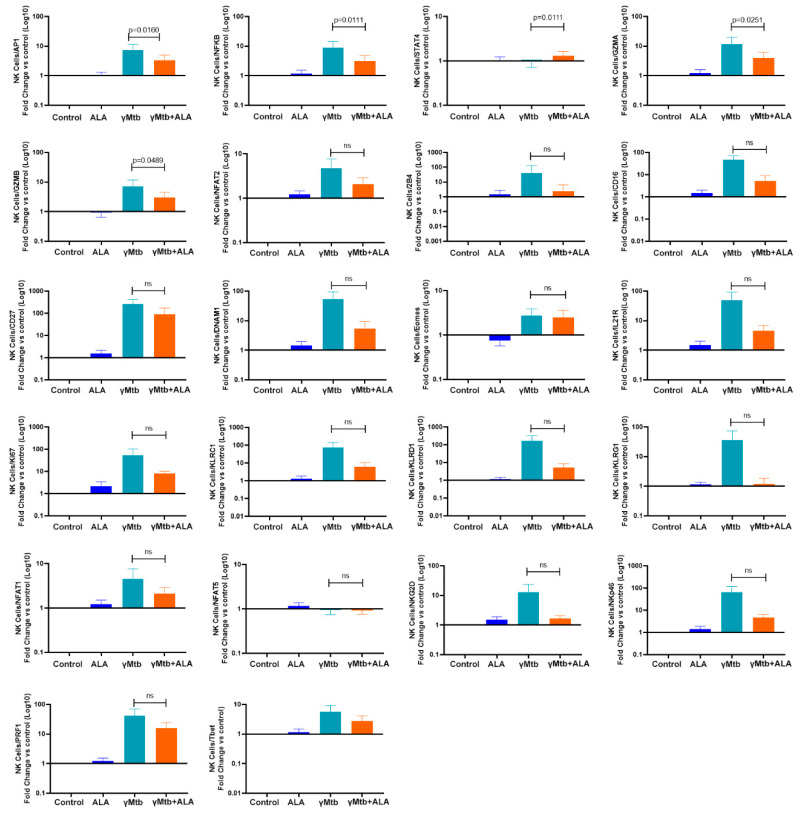
ALA inhibits NF-κB, AP1, GZMA, and GZMB gene expression in γ*Mtb*-cultured NK cells. PBMCs of healthy LTBI+ donors were cultured with or without γ*M*tb (10 μg/mL) and in the presence or absence of ALA, as mentioned in the Methods section. After 48 h, NK cells were isolated by flow sorting, RNA was collected, and real-time PCR analysis was performed to determine various signaling molecules and transcription factors. PBMCs from six LTBI+ donors were used for this experiment. Paired *t*-tests were used to compare the differences between untreated and treated PBMCs from the same samples. The significant *p* values are shown (*p* < 0.05); ns: not significant, *p* > 0.05.

**Figure 6 ijms-24-07267-f006:**
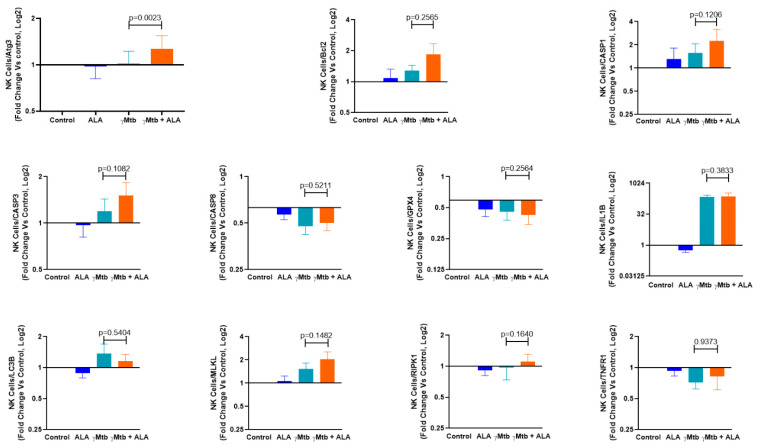
ALA inhibits antimicrobial peptide expression and enhances Atg3 expression in γ*Mtb* cultured NK cells. PBMCs of healthy LTBI+ donors were cultured with or without γ*Mtb* (10 μg/mL) and in the presence or absence of ALA, as mentioned in the Methods section. After 48 h, NK cells were isolated via flow sorting, RNA was collected, and real-time PCR analysis was performed to determine various death pathway gene expressions. PBMCs from six LTBI+ donors were used for this experiment. Paired *t*-tests were used to compare the differences between untreated and treated PBMCs from the same samples. The *p* values are shown in the figure.

**Figure 7 ijms-24-07267-f007:**
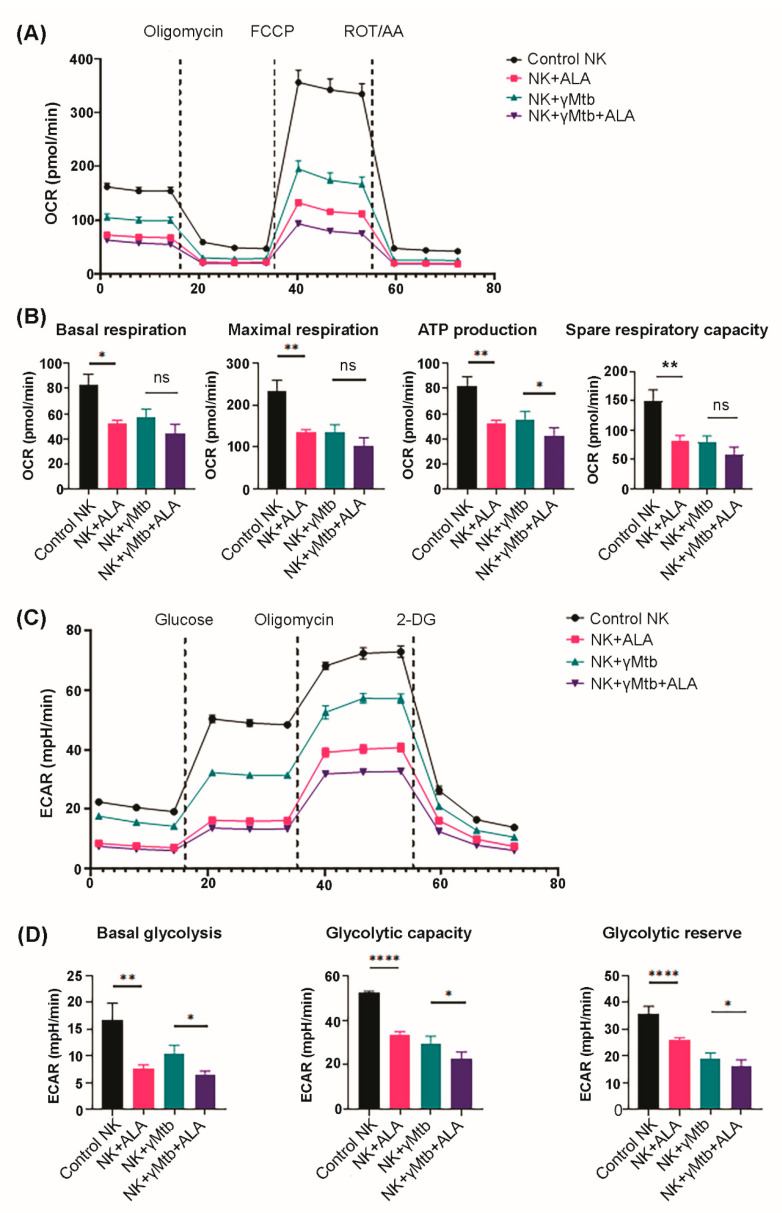
ALA treatment switches NK cells to an energetically quiescent state. PBMCs of healthy LTBI+ donors were cultured with or without γ*Mtb* (10 μg/mL) and in the presence or absence of ALA (3 µM), as mentioned in the Methods section. After 48 h, NK cells were isolated via magnetic cell sorting and subjected to extracellular flux analysis using an Agilent Seahorse XFe96 analyzer. NK cell glycolysis was measured with the sequential addition of glucose, oligomycin, and 2-DG. Similarly, OXPHOS parameters were measured in isolated NK cells after the addition of oligomycin, FCCP, and rotenone/antimycin: (**A**,**B**) mitochondrial OCR and (**C**,**D**) ECAR were measured; graphs (**A**,**C**) show mitochondrial OCR (**A**) and ECAR (**C**) in real time as kinetic graphs; graph (**B**) shows the collective OXPHOS parameters of basal respiration, maximum respiration, ATP production, and spare respiratory capacity as bar graphs. The *p* values were derived using an unpaired 2-tailed independent *t*-test. The mean values and SEMs are shown; (**D**) bar graphs show the collective glycolytic parameters such as basal glycolysis, glycolytic capacity, and glycolytic reserve. In (**B**,**D**), for all panels, the data are presented as mean ± SEM (*n* = 12; 4 statistical replicates from 3 individual donors); each parameter between treatments was compared using independent Student’s *t*-test: * *p* < 0.05, ** *p* < 0.01, and **** *p* < 0.0001. ns: not significant.

**Figure 8 ijms-24-07267-f008:**
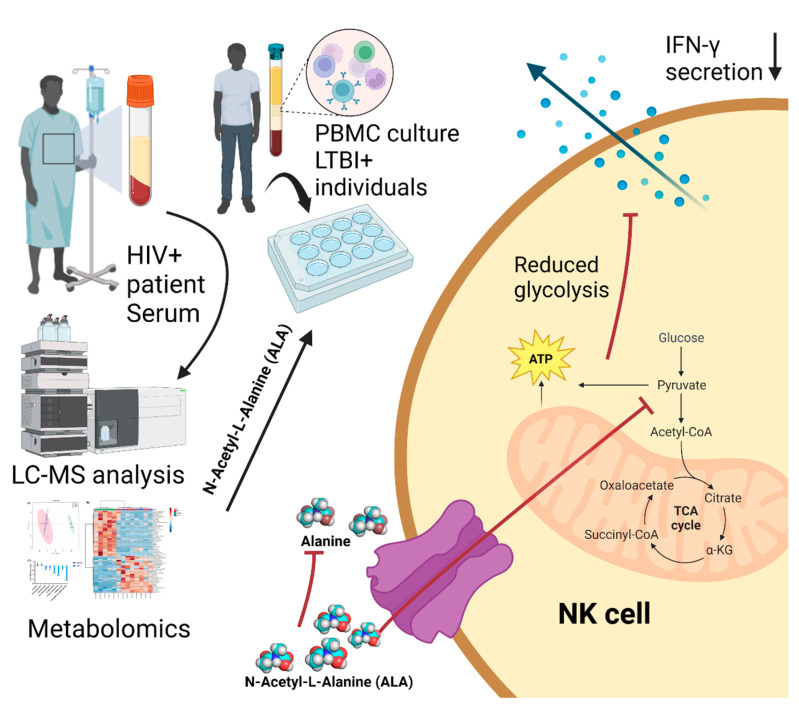
Schematic representation of the potential mechanism of how ALA regulates the immune response of the PBMCs from the LTBI+ individuals. The HIV-differentiated metabolite ALA targets NK cells and functions as a downregulator to reduce NK cell glycolysis, resulting in NK cell autophagy and the reduction in IFN-γ production.

## Data Availability

All data supporting the findings of this study are available in the manuscript. If there are any special requests or questions for the data, please contact the corresponding author (G.Y.).

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
