# Peer review of "HIV-Differentiated Metabolite N-Acetyl-L-Alanine Dysregulates Human Natural Killer Cell Responses to Mycobacterium tuberculosis Infection"

_ijms, 2023, doi:10.3390/ijms24087267_

Round 1

Reviewer 1 Report

The increase in the number of persons with HIV infection identified at Stages 4B—5 hampers the diagnosis and reduces the efficiency of TB treatment, leading to a severe course of the latter and death on average within the first year after diagnosis. It is necessary to improve the system for control, prevention and organization of medicalcare to both TB and HIV-infected patients, by implementing the best medical practices globally and the WHO recommendations. The study of the immunological mechanisms mediating the mutual influence of tuberculosis and HIV contributes to the creation of new means of combating these deadly infections.

I would like to make the following remarks:

1. WHO estimate of TB mortality in 2021 is 1.567 millions both HIV-positive and HIV-negative (https://worldhealthorg.shinyapps.io/tb_profiles/?_inputs_&lan=%22EN%22&entity_type=%22group%22&group_code=%22global%22). Authors' reference to the paper published in 2010 is outdated.

2. The probands must be characterized in more detail. Age, sex, criteria of inclusion/exclusion, how LTBI was detected? For how long? Details of ART, etc. Otherwise it is not clear if the groups of probands are comparable.

3. In Materials and methods, Statistics, I do not understand the phrase (line 181): "the experiments were repeated at least once to ensure reproducibility"?

4. Why one needs both bars and dots on Figures 2,3,4,5,6,7? I would suggest to leave bars and save dots.

5. Figure 2 is absolutely unreadable. 

Author Response

Response to Reviewers’ comments

Response to Reviewer #1

  1. WHO estimate of TB mortality in 2021 is 1.567 millions both HIV-positive and HIV-negative (https://worldhealthorg.shinyapps.io/tb_profiles/?_inputs_&lan=%22EN%22&entity_type=%22group%22&group_code=%22global%22). Authors' reference to the paper published in 2010 is outdated.

We thank the reviewer for pointing out that the data we cited is outdated. Now we updated with the latest data from WHO according to the reviewer’s suggestion, and added the new reference accordingly.

  1. The probands must be characterized in more detail. Age, sex, criteria of inclusion/exclusion, how LTBI was detected? For how long? Details of ART, etc. Otherwise it is not clear if the groups of probands are comparable.

We agree with the reviewer that those information of the probands are important. Now that we added the detailed information of the probands as shown in a new table (Supplementary Table 1). We also added two paragraphs in the methods section with related references to further address the reviewer’s concern.

  1. In Materials and methods, Statistics, I do not understand the phrase (line 181): "the experiments were repeated at least once to ensure reproducibility"?

We regret for the confusion. Now that we changed it to “the experiments were repeated at least 1-2 times to ensure reproducibility".

  1. Why one needs both bars and dots on Figures 2,3,4,5,6,7? I would suggest to leave bars and save dots.

We agree with the reviewer, and remade all figures with only using the bar plot. We also made the resolution quality of the figures much better than the first version.

  1. Figure 2 is absolutely unreadable. 

We agree with the reviewer, and remade the Figure 2.

Reviewer 2 Report

In this work, the authors have studied the interactions of Mycobacterium tuberculosis along with HIV. Mycobacterium tuberculosis is a global threat that causes about 2 million death every year. One third of the world population is estimated to be carrier of this pathogen. The pathogen can remain dormant for years inside the host before its emergence in an immunocompromised state. HIV on the other hand can lead to decrease immune response leading to increased chance of active tuberculosis. The authors have addressed an important question in the field regarding the mechanism of HIV in dysregulate immune responses s in 30 LTBI+ individuals. Using genetic and metabolomics tools the authors have found that six metabolites were significantly less abundant, and two were significantly higher in abundance in HIV+ individuals compared to healthy donors: N-Acetyl-L-Alanine (ALA), inhibits pro-inflammatory cytokine IFN- É£ production by NK cells of LTBI+ individuals. ALA inhibits glycolysis of LTBI+ 42 individuals' NK cells in response to Mtb, thereby shedding light on the interaction between the etiological agents of tuberculosis and AIDS.

Overall, the work is good and addressed an important area of Tuberculosis research. I’ve two minor comments I’m listing below:

1.Figure 1 heat map is legends are hard to read. Please change it with a better-quality image.

2. The authors should provide a concluding or discussion image where they can summarize their finding.

Author Response

Response to Reviewer #2

1.Figure 1 heat map is legends are hard to read. Please change it with a better-quality image.

We are sorry for the low-resolution Figure 1 in our 1st version. Now that we remade the figure and increased the quality and resolution, now the heat map is readable.

  1. The authors should provide a concluding or discussion image where they can summarize their finding.

We thank the reviewer for the constructive suggestion, which will make the audience much easier to understand out findings. We have made the Figure 8 to summarize opur findings.

Round 2

Reviewer 1 Report

The authors made a good job improving the text and the results presentation.

Still the information on probands is incomplete. As the authors are discussing metabolic parameters, the groups of probands should be matched also in sex and age.

And I still do not understand reproducibilty analysis with 1-2 repeats (line 193).  

Author Response

Responses to the reviewers’ comments

#Reviewer 2:

The authors made a good job improving the text and the results presentation.

We appreciate the reviewer’s compliment, and we thank the reviewer again for the comments that helped improve our manuscript.

Still the information on probands is incomplete. As the authors are discussing metabolic parameters, the groups of probands should be matched also in sex and age.

Thank the reviewer for the comment again on the probands description. We respectfully differ from the reviewer’s comment on the above. We believe that finding a common metabolite marker for all HIV-infected patients, beyond the patient age, sex, and ART-treatment, etc is more significant in clinical settings. Our metabolomics analysis also demonstrated that the HIV patients (either treatment naïve or ART-treated, actually with different age and sex) show similar metabolic signatures, and our metabolite marker ALA are tested and verified based on the metabolite analysis. As such, in the inclusion/exclusion criteria, the enrollment included both sex and an age range 20-60 years. We believe this criteria is of more clinical significance. We hope our explanation can address the reviewer’s concern.

And I still do not understand reproducibility analysis with 1-2 repeats (line 193).  
